# Systematic Review of Fault Tolerant Techniques in Underwater Sensor Networks

**DOI:** 10.3390/s21093264

**Published:** 2021-05-08

**Authors:** Lauri Vihman, Maarja Kruusmaa, Jaan Raik

**Affiliations:** 1Smart City Center of Excellence (Finest Twins), Tallinn University of Technology, 12618 Tallinn, Estonia; jaan.raik@taltech.ee; 2Department of Computer Systems, Tallinn University of Technology, 12618 Tallinn, Estonia; maarja.kruusmaa@taltech.ee

**Keywords:** underwater sensor network, fault tolerance, cross-layer fault tolerance, fault management

## Abstract

Sensor networks provide services to a broad range of applications ranging from intelligence service surveillance to weather forecasting. While most of the sensor networks are terrestrial, Underwater Sensor Networks (USN) are an emerging area. One of the unavoidable and increasing challenges for modern USN technology is tolerating faults, i.e., accepting that hardware is imperfect, and coping with it. Fault Tolerance tends to have more impact in underwater than in terrestrial environment as the latter is generally more forgiving. Moreover, reaching the malfunctioning devices for replacement and maintenance under water is harder and more costly. The current paper is the first to provide an overview of fault-tolerant, particularly cross-layer fault-tolerant, techniques in USNs. In the paper, we present a systematic survey of the techniques, introduce a taxonomy of the Fault Tolerance tasks, present a categorized list of articles, and list the open research issues within the area.

## 1. Introduction

Underwater Sensor Networks (USNs) have become widespread and are being deployed in a wide range of applications ranging from harbor security to monitoring underwater pipelines and fish farms. Due to the fact that USNs often operate in an extremely harsh environment, and many of their applications are safety-critical, it is imperative to develop techniques enabling these networks to tolerate faults. Moreover, USNs face many challenges that are not present in terrestrial networks, such as virtual inapplicability of the wireless radio communication under water and limitations of the acoustic means, for example.

In the current paper, applications, practices, and central issues on fault tolerant USNs are discussed, and a systematic survey of fault tolerant techniques in USN networks is presented. Our objective is to investigate the state of the art and main focuses of ongoing research on cross-layer Fault Tolerance in underwater sensor networks, as well as to identify the existing gaps in previous research. As by now a limited effort has been put on the Fault Tolerance of USNs by the research community, the criteria is expanded, and papers covering some specific aspects of the fault-tolerance topic are also taken into account. Moreover, the sources also include generic terrestrial Fault Tolerance in sensor networks because research on underwater sensor network faults is limited, and many of the generic technologies, approaches, and tools can be adapted for use in USNs.

It is important to stress that the underwater environment is mostly different from terrestrial conditions, in the sense of additional and more fatal hazards, like an increased pressure and a danger of flooding, as well as added difficulty of communication and physical access. Some communication media, such as radio signals, are not applicable underwater. Additionally, falling temperatures with increasing depth may affect the equipment’s operation and reliability.

In this paper, a systematic search in IEEExplore, Google Scholar, and ScienceDirect online environments was carried out to obtain a relevant sample of works in the field of fault tolerant techniques in USNs. The search revealed 122 papers, with 59 of them dedicated to the Fault Tolerance of USNs, while the 63 remaining ones were generic fault tolerant techniques for terrestrial sensor networks applicable to the underwater environment.

In order to provide a systematic view of the paper categories, this survey introduces a taxonomy of Fault Tolerance tasks. Specifically, the identified relevant papers are grouped according to the tasks of fault prevention and prediction and Fault Detection and Fault Identification, as well as Fault Isolation and Fault Masking, respectively.

Moreover, a comparative analysis of the identified papers was presented, where the works were characterized according to their extra-functional aspects covered (i.e., security, energy-efficiency, scalability, cross-layer aspect) and Fault Tolerance tasks targeted, as well as marine or terrestrial application. As a result of the analysis, the lack of cross-layer Fault Tolerance approaches in the USN domain was identified as a particular gap in the state-of-the-art with prospective future research.

There are several surveys investigating underwater sensor networks. For example, Reference [1] introduced the term of Internet of Underwater Things (IoUT) and showed its applications in fish farms, monitoring underwater pipelines, harbor security, etc., and [2] analyzed cross-layer error control in Underwater Wireless Sensor Networks (UWSNs); however, the analysis focused on the underwater wireless network functionality faults and not on other sources of the USN faults. Underwater communications have been specifically surveyed in References [3,4], disregarding aspects of underwater sensor networks outside communication issues.

The main challenges identified for Internet of Underwater Things are the communication reliability and the differences between Underwater and Terrestrial Networks [5], such as mobility caused by water flow. For terrestrial sensor networks, there were 11 surveys found. Thereof, 3 terrestrial surveys addressed cross-layer aspects. Reference [6] was surveying cross-layer resilience design methods and [7,8] fault management techniques in wireless sensor networks. In addition, References [9,10] included surveys about aspects of the internet of things, and 7 papers by References [7,8,11,12,13,14,15] were surveys of different aspects of terrestrial wireless sensor networks. Reference [16] presented a survey about fault tolerant control systems, and, finally, Reference [14] was focused on surveying fault management frameworks in terrestrial wireless sensor networks.

The current state-of-the-art is lacking literature reviews covering faults in USNs not only from communication but from the entire infrastructure perspective, as well. To that end, the current paper has the following novel contributions:to the best of the authors’ knowledge, this is the first survey of fault-tolerant, particularly cross-layer fault-tolerant, techniques in USNs;it introduces a taxonomy of the Fault Tolerance tasks for categorizing fault-tolerant techniques for USNs;it presents a comprehensive, categorized list of articles of works applicable in fault-tolerant USN design and deployment; andthe survey also lists the open research issues within the focused area.

The paper is organized as follows. In Section 2, the formal methodology of selecting the papers is explained and a breakdown of the sample by keywords is provided. Section 3 gives an overview of the specific Fault Tolerance challenges in underwater sensor networks. In Section 4, the taxonomy of possible fault sources and that of Fault Tolerance tasks is presented. Subsequently, Section 5 is divided according to this taxonomy of tasks. In Section 5.1, works targeting the fault prevention and prediction task are discussed and the respective design, deployment, data collection, and testing frameworks are reviewed. Section 5.2 gives an overview of Fault Detection and Fault Identification techniques. Section 5.3 provides an overview of Fault Masking and Fault Recovery techniques. In Section 6, a categorized table of the related works identified by the survey is presented. Finally, in Section 7, open research issues are discussed, and conclusions are drawn in Section 8.

## 2. Methodology

The current overview is following the PRISMA [17] guidelines for systematic reviews. In order to obtain a relevant sample in the field of fault tolerant techniques in USNs, IEEExplore, Google Scholar, and ScienceDirect online environments were searched with the following search keywords: “underwater”, “sensor network”, “internet of things”, “resilient”, “fault tolerant”, “fault management”, “cross-layer” in English language. Because the resulting counts were low (see Table 1), some keywords were removed, and more papers identified. Top papers were selected by the order of relevance offered by the respective environments. The papers published before the year 1990 were not considered. Further, citations within those sources were searched from the aforementioned environments, and additional papers were identified this way. Related articles offered by IEEExplore and ScienceDirect algorithms were also taken into account. Next, the duplicates and non-relevant papers (e.g., control theory) were removed from the collected papers, and the collected papers were analyzed, categorized, and divided into marine and terrestrial categories. Personalization on search engines was turned off wherever possible.

Table 1 shows the count of results using combinations of keywords in Google Scholar, IEEExplore, and Sciencedirect. (Searches were conducted on 13 April 2021, from Taltech, Estonia, IP addresses.). From Table 1, it can be seen that some combinations were giving no, or a very limited number of, results. A critical amount of papers was not reached using the initial criteria, and the criteria were expanded to include also relevant non-marine-specific (terrestrial) papers. The argumentation behind this is that many of these techniques may also be usable in underwater environments (see Section 3).

As a result of the search procedure, 122 related works were identified. These included 59 papers on marine Fault Tolerance and 63 papers being on terrestrial. The papers were tagged by specific areas addressed by them. The tags for specific areas included ’sensor network’, ’fault tolerant’, ’wireless’, ’scalable’, ’mobile’, ’routing protocol’, ’security’, ’localization’, ’framework’, ’survey’, ’energy-efficient’, ’cross-layer’, ’deployment’, ’marine’, and ’terrestrial’.

A bar graph showing the number of papers from our search that covered different specific areas is presented in Figure 1. The specific areas are ordered by the number of papers addressing them, and the bars for the specific areas maintain their colors throughout Figure 1, Figure 2 and Figure 3. It should be noted that, in the following context, the meaning of “localization” is location detection in space, and the meaning of “mobile” is capacity of movement. It can be seen from Figure 1 that there were substantially more terrestrial papers than the ones specific to marine environments. In addition, wireless communication is a frequently targeted area. Figure 2 shows research areas of the analyzed papers falling into terrestrial category. It should also be noted that papers on general fault-tolerant sensor networks, not specifically claiming any environments, were categorized into the terrestrial category. Figure 2 presents the frequency of specific areas addressed in terrestrial papers where the order of the most frequent categories has switched but is not much different from Figure 1. However, Figure 3, which presents the analyzed marine and aquatic environment-related papers covering different specific areas, shows that marine wireless communication related research works have the highest number of papers among those identified by the current survey.

In order to further highlight the differences of the previous research focus in marine and terrestrial sensor networks, a radar diagram is shown in Figure 4. For the diagram, we selected eight significant specific areas: ’fault tolerant’, ’wireless’, ’mobile’, ’localization’, secure’, ’scalable’, ’energy efficient’, and ’cross-layer’, respectively. It can be seen from Figure 4 that a large share of marine research (shown by blue color in Figure 4) interest from the identified papers has been drawn to underwater wireless communication, while some are drawn to underwater Fault Tolerance techniques and almost none to underwater cross-layer Fault Tolerance. Underwater energy-efficiency and scalability are more covered areas than underwater vehicles (mobility) and security. Papers addressing terrestrial techniques (shown by green in Figure 4) were, according to the initial search criteria, more focusing on Fault Tolerance, including cross-layer Fault Tolerance, and less on energy efficiency or security.

High research effort on marine wireless networking in Figure 4 confirms the claim [5] that current pace of research on Internet of Underwater Things (IoUT) is slow due to the challenges arising from the uniqueness of underwater wireless sensor networks. Specifically, the main challenges for IoUT are the differences between Underwater Wireless Sensor Networks and Terrestrial Wireless Sensor Networks [5].

Fault Tolerant Control Systems is another extensive research area of Fault Tolerance not covered by current paper. There is an existing recent review paper [16] on the overview of research works in that topic.

## 3. Specifics of Underwater Sensor Networks

Environmental and engineering challenges for sensor networks in underwater environments are shown on Figure 5. An underwater environment is mostly different from a terrestrial one due to the harsh physical conditions—high pressure and hard accessibility, as well as limited communication and energy resources. Depending on the specific location, the temperature may fall with increasing depth, which may affect, e.g., the battery lifetime. In underwater environments, faults can be caused over time by ambient flowing water generated by surface waves or other reasons that shake the components of the sensor networks. Moreover, faults can be introduced by humans or aquatic organisms.

Many communication methods are unavailable underwater, and there are multiple phenomena [2,18] that obstruct communication there. Because of the possibility of flooding the hardware due to water leakage, more attention and resources should be paid to the physical integrity of sensor nodes. On the other hand, faults from excessive heat should be rare and avoidable underwater. In the underwater context, Fault Tolerance has been so far addressed for reliant UWSN networking [2,3,19,20], space localization [21], and monitoring underwater pipelines [22]. While it should be possible to adapt most of the generic Fault Tolerance concepts for the underwater use, the environment is more demanding and unforgiving, and faults are more costly. Some more demanding approaches, like cloud computing, may not make sense to be implemented in USNs. However, the authors cannot see any obstacles for applying those fault tolerant approaches that yield appropriate communication methods, low network bandwidths, and power requirements in the underwater domain.

Last but not least, one of the promising approaches that could be adapted successfully within the underwater environment’s constraints appears to be cross-layer resilience, which is an open research topic and lacking in recent research works, even for the terrestrial implementations.

## 4. Taxonomy of Faults and Fault Tolerance Tasks

In the following, we present the taxonomy of the sources of faults, as well as of the Fault Tolerance tasks. The objective of describing and representing these taxonomies is to categorize the articles for the current survey.

### 4.1. Sources of Faults

A fault is defined [23] as an underlying defect of a system that leads to an error. An error is a faulty system state, which may lead to failure, and failure is an error that affects system functionality. Faults may occur in different components and layers of systems for different reasons. The only type of fault possible in software is a design fault introduced during the software development, i.e., a bug [24]. Software bugs can be addressed separately and will not be covered further in the current paper.

Fault sources can be categorized by components where they occur. In sensor networks, they can occur in sensor nodes, in the communication network, and in the data sink [25]. Sensor networks share common failure issues with traditional networks, as well as introduce node failures as new fault sources [7].

USNs additionally introduce faults caused by environmental conditions, such as pressure, currents, underwater obstacles, etc. Those conditions may cause physical damage that may result in failures, as well as obstruct the system’s functionality. For instance, in underwater acoustic networks, loss of connection and high bit error rate may be caused by shadow zones [18] formed by different physical reasons. Domingo and Vuran distinguish up to five different underwater propagation phenomena which may obstruct communication [2].

Faults can either be permanent or temporary [26]. Permanent faults may be caused by manufacturing defects, as variances of the hardware components are inevitable due to physical reasons [27]. One of the other factors that can introduce faults is aging and wear-out of the hardware components [28]. In addition to the components themselves, the interconnections between them are also affecting the reliability and may cause faults [29].

One of the challenges of fault management is temporary faults, especially soft errors. Soft error is a temporary change of signal value due to ionizing particles [26] that may lead to failure. Due to high integration density, it is estimated that soft failure rate is increasing in the future [30]. Another potential source of temporary faults is electromagnetic interference [31].

### 4.2. Fault Tolerance Tasks

The objective of the current section is to define a taxonomy of Fault Tolerance tasks to help categorize the identified papers. The Fault Tolerance tasks are based on more general Fault Tolerance principles from References [32,33]. Figure 6. shows the taxonomy of Fault Tolerance tasks applicable in USNs and how they affect each other. While the design and initial deployment of USNs contribute to Fault Prevention and Prediction abilities, data collecting techniques at the run-time contribute also to Fault Detection and Fault Recovery stages of the system, all of which are going to be discussed in the current paper.

The techniques under consideration can be categorized into the following groups:Fault Prediction and PreventionThis task is about both preventing a fault to happen, as well as about proactive fault avoidance. Sensor networks can prevent certain faults from happening by design and/or deployment aspects. A disadvantage of fault prevention is a potentially increased system complexity. Fault avoidance, in turn, includes manufacturing testing and verification, which have a high cost often exceeding that of the entire design process.Fault Detection and IdentificationOne of the central parts of Fault Tolerance is Fault Detection and Fault Identification of affected components which can, for instance, be performed by utilizing data collection with ping messages. Without Fault Identification, for instance, sensor node and network faults may be hard to distinguish. A disadvantage of Fault Detection and Fault Identification may be additional energy requirements and network congestion.Fault Isolation, Masking, and RecoveryIsolation, masking, and recovery are different techniques for repairing a fault, minimizing the effect of a fault, or avoiding it to turn to system failure. Identified faults can be isolated, masked, and sensor network recovered, for instance, redirecting traffic through healthy backup components. Fault Recovery can ensure overall system operation even when components degrade. The downside may be the cost of adding components to ensure redundancy.

The overview of fault tolerant techniques presented in the following section follows the above-described taxonomy.

## 5. Overview of Techniques by Fault Tolerance Tasks

In the following, the Fault Tolerance techniques categorized according to the Fault Tolerance Tasks introduced in Section 4.2 and presented in Figure 6 will be discussed in more detail.

### 5.1. Fault Prevention and Prediction

Fault prevention and prediction in sensor networks are dependent on the architectural design of the system and the initial deployment method of the sensor network. These will be discussed in the following subsections. In addition, data collection in USNs and testing frameworks for UWSNs are presented.

#### 5.1.1. Design of the Sensor Network

In Wireless Sensor Networks (WSN), instead of a centralized homogeneous topology, dividing nodes into clusters is an energy efficient and resilient method [12], where dedicated cluster head nodes may have more energy and communication capabilities to effectively act as mediators between regular nodes and data sinks.

To overcome the issues caused by varying environmental challenges of Underwater Wireless Sensor Networks (UWSN), natural algorithms may be utilized. For instance, clustering and routing can be done utilizing Cuckoo Search algorithm and Particle Swarm Optimization [34], which have behaved more resiliently in underwater conditions than more usual terrestrial Low Energy Adaptive Clustering Hierarchy (LEACH) protocol [11]. Pressure measurements have been used for UWSN routing [35] with floating depth-controlling sensors. Fault Management tasks can also be distributed across the whole network. In WSN with enough spare nodes energy efficient grid can be formed [36], changing the node manager, gateway and sensing nodes selected and spare nodes put to sleep. This results in energy-efficient and lightweight network but requires excess nodes.

However, existing UWSN protocols have not been adequately compared in underwater field trials yet [4].

#### 5.1.2. Sensor Network Deployment

Sensor network deployment techniques are important for WSNs where deployment may directly affect the nodes’ locations and networking availability. Even for terrestrial wireless sensor networks, to obtain a satisfactory network performance, an adaptable deployment method is essential [37]. Usually, the sensor placement for WSNs utilizes, for redundancy reasons, more sensors than the minimum required number [38]. The deployment costs and energy efficiency of WSNs have been investigated in Reference [39], and it has been found that there is no single solution that can easily be applied in practice [40].

Wired sensor network deployment is less researched, possibly because wired sensor networks’ node deployment locations are limited by the cables, their locations are more predetermined, and node connectivity is not directly related to the location.

#### 5.1.3. Data Collection

Sensor networks tend to have limited network bandwidth, energy, and storage capabilities. Thus, filtering and aggregating sensor information may be a way to meet those requirements. Raw sensor data near the source can be divided into informative, non-informative, and outlier groups [41], and only the needed data could be communicated or stored. Outlier data may result from noise, failures, disturbances, etc., and may be useful for Fault Tolerance purposes.

Different techniques to compress and aggregate collected information in UWSNs are investigated in Reference [42]. It was found that aggregation is justified, and cluster-based aggregation techniques are performing better than non-cluster-based ones. For instance, cluster head (CH) switching to backup (BCH) technique was proposed [43] for cluster-based UWSNs.

Moreover, security challenges need to be addressed. One way to minimize the risk of data tampering and/or interference is to ensure that the data is processed locally or, if that is not possible, then communicated end-to-end encrypted [44].

#### 5.1.4. UWSN Testing Frameworks

Wireless networking protocols are one of the key research areas in UWSNs. To evaluate the implementation of underwater wireless protocols, simulation is often used. Due to the specifics of underwater environments (See Section 3), generic simulation environments are not able to capture some of the relevant features. Frameworks covered in the current section are useful for underwater acoustic protocols’ simulation and evaluation.

Frameworks, such as DESERT version 1 and 2 [45] and SUNSET [46], that allow simulation, emulation, and testing of the sensor networks, have been developed for UWSNs. An analysis conducted in Reference [47] shows that SUNSET represents a more mature, flexible, and robust framework for in-field testing than DESERT. However, DESERT v2 was released subsequently. For acoustic UWSN security testing, SecFUN framework [48] has been proposed.

### 5.2. Fault Detection and Identification

In essence, Fault Detection means determining that one or more bits in the computation differ from their correct value [33]. This can be detected via continuous monitoring of the network and nodes’ status. Some sources also use the word “Diagnosis” in a broader meaning than just detection and identification. Diagnosis has been defined as “characterizing the system’s state to locate the causes of errors, determine how the system is changing over time, and predict errors before they occur [33]”. The current section covers different techniques to execute the previously mentioned concepts.

A distributed hierarchical fault management [49] has been used for WSNs, where agent Fault Detection devices collect information from the power modules and sensors to determine failure conditions and sequentially diagnose the nature of the detected failure.

At higher abstraction levels, there has been a wide use of the SNMP protocol [50] by the industry for Fault Detection querying and triggering in IP networked devices. There are multiple commercial tools for generating failures, e.g., Chaos Monkey from Netflix [51], that randomly terminate services in production environments, to ensure their resiliency. The latter does not manage the occurring faults but ensures that the repairing mechanisms are in place and operable. Intelligent Platform Management Interface (IPMI) [52] is an industrial technology specification for hardware system management and monitoring.

A neural-network-based scheme for sensor failure detection, identification, and accommodation can be used which may allow the conditions to deviate to greater extent from theoretical models and estimation. A relatively simple and computationally light approach has been presented [53], where a neural network is used as an online learning state estimator for detecting faults. The neural network itself can be built as fault-tolerant [54], so that failing nodes have the least impact on result data.

Situational Awareness approach, using a mechanism that has been borrowed from humans, can be applied in sensor data interpretation for Internet of Things (IoT), specifically, regarding processes of sensation, perception and cognition. In addition to specification-based and learning-based approaches, a perception-based approach utilizing Fuzzy Formal Concept was proposed [55] for Situational Awareness identification.

Semantic Sensor Network Ontology has been proposed in Reference [56] for managing interoperability between sensing systems. The Semantic Ground describes information for interoperability and cooperation among agents [57]. To enhance resilience in Semantic Sensor Networks, monitoring nodes may forward observations to association nodes, which develop Situational Awareness by mining association rules, for example, via a natural Artificial Bee Colony algorithm [57].

Electric Power Grids need efficient monitoring since, for outage detection, environmental monitoring, and fault diagnostics, different WSN-based approaches are reviewed [13]. Most of these approaches are also applicable in other kinds of applications.

### 5.3. Fault Isolation, Masking and Recovery

Subsequent to Fault Detection, Fault Identification, and Fault Diagnosis, a fault handling stage can be entered [49] to prevent further data corruption and system deterioration. The fault handling consists of Fault Isolation, Masking, and Recovery. Fault handling can hide the fault occurrence from other components by applying Fault Masking; the key techniques for such masking are informational, time, and physical redundancy [32]. Proposed masking technique For Underwater Vehicles is Triple Modular Redundancy (TMPR) [58], which is also one of the most commonly used Fault Masking techniques. Isolating a faulty component from the others can be facilitated by using virtualization [32]. In large scale distributed systems, frozen virtual images of healthy services have been used as checkpoints [59] for rolling back in case of a fault occurrence.

Fault Recovery ensures that the fault does not propagate to visible results, for instance, by rolling back to a previous healthy state (checkpointing) or re-trying failed operations (time redundancy). Some of the techniques for Fault Recovery can be Reconfiguration, which is changing the system’s state so that the same or similar error is prevented from occurring again, and Adaptation, which is re-optimizing the system, for instance, after Reconfiguration task [33].

In Sensor Networks, different approaches for Fault Recovery have been used, that have different resource overheads, energy-efficiencies, scalabilities and network types. For both network and node Fault Recovery in wireless sensor networks, Mitra et al. (2016) [8] compares techniques, such as checkpoint-based recovery (CRAFT), agent-based recovery (ABSR), fault node recovery (FNR), cluster-based and hierarchical fault management (CHFM), and Failure Node Detection and Recovery algorithm (FNDRA). While some of those are specific to terrestrial wireless usage, some principles (e.g., checkpointing, etc.) can also be used in wired and/or underwater environments. To reduce the network bandwidth requirements, checkpoint backup can be mobile to nearby nodes [60] and used for recovering from fault situations.

In network protocols, Fault Masking and Fault Recovery are handled by error control schemes that are commonly categorized into the following three groups [2]:Automatic Repeat Request (ARQ)—re-transmission of corrupted data is asked;Forward Error Correction (FEC)—data corruption can be detected and corrected by the receiving end; andHybrid ARQ (HARQ)—a combination of FEC and ARQ.

The cross-layer approach benefits Fault Recovery significantly since single-layer redundancy, such as hardware redundancy and application checkpointing, have very high costs, and latency between fault occurrence and detection makes the recovery difficult [33].

## 6. Comparative Analysis

All the papers that were selected according to the criteria described in Section 2 are listed in Table 2. The table includes information about the targeted extra-functional aspects and Fault Tolerance task(s). In addition, the Marine column in Table shows if the listed paper is explicitly touching aquatic environments. The papers are ordered by their order of citation within this survey paper. Papers that are not directly cited in the text but still listed in Table 2 are ordered chronologically by the publishing year. Papers that are not included in the analysis but are cited (e.g., definitions) have not been included in the table.

It can be seen from Table 2 that only two papers address both marine and cross-layer Fault Tolerance aspects. However, in the work targeting cross-layer analysis of error control [2], the term ’cross-layer’ does not apply to the system stack but only to the communication protocol layers. Another work authored by the authors of this survey [61] is focusing on data-driven cross-layer Fault Tolerance. Thus, there is a serious gap in research addressing cross-layer Fault Tolerance in underwater sensor networks.

Regarding other extra-functional aspects, security in marine environments is addressed by six marine papers and is focusing on securing wireless communication [20,48,62], authentication [63], and hybrid attacks [64]. On scalability, seven marine papers were identified, and underwater scalability has been researched, for instance, in the context of monitoring underwater pipelines [22]. On Energy-efficiency, there were 14 Marine papers identified, and extensive focus has been on energy-efficient underwater wireless protocols [3,19,65,66,67,68,69] and less on other aspects. Open research issues from all the mentioned extra-functional aspects will be discussed in the following section.

## 7. Open Research Issues

In the following, the open research issues identified are presented according to the categories of extra-functional aspects reported in Table 2.

### 7.1. Security

Faults and security are interrelated concepts [59]. It requires effort to prevent systems from being penetrated, even when they operate as intended; however, faults will add further uncertainty and make the task of prevention even harder. Faults can be created by an intrusion; but, moreover, faults can enable new intrusion vectors [70]—misbehaving devices violate key assumptions and create a number of new attack vectors to systems. For example, soft errors explained in Section 4.1 can be used to defeat cryptography [128]. In wireless sensor networks, intrusion detection systems have been investigated [71], and intrusion detection can be divided into Anomaly detection, which can work well for unknown attacks, and Misuse detection, for known attack signatures.

### 7.2. Energy-Efficiency

Power dissipation has by now reached a point where energy concerns limit the computation we can deploy on the chip [70], and the aim is shifting from transistor density and speed to energy density and cost. Energy density and efficiency need also to be addressed on a larger scale; for instance, WSNs may not have unlimited power supply and need to utilize energy-efficiency strategies [11,12,36,40]. For Fault Tolerance techniques, cross-layer approach is considered more energy-efficient [33] than single layer. Strategic redundancy in cross-layer approach may allow systems to safely operate on the verge of failure [70], spending less energy without going over the edge.

In sensor networks, energy consumption can be reduced, for instance, by using specific low-energy communication protocols, reducing the number and speed of the nodes, and pausing the nodes [129]. However, with the growing complexity of applications, energy consumption is becoming one of the limiting factors.

### 7.3. Scalability

One of the traditional benefits of scaling has been the decrease of cost per functionality [70], but easing reliability problems by multiplicating logic, voting and similar techniques means that the scaled technology might not offer a reduction of energy or area. Some Fault Tolerance techniques may increase computing overhead, and not all approaches are scalable [8]. Large scale fault tolerant systems are researched without paying special attention to energy and communication constraints [59].

### 7.4. Cross Layer Approach to Fault Tolerance

Faults are not going to disappear but likely to increase in the future [30]. One way to cope with faults is to accept imperfect devices to fail and compensate failures at higher levels in the system stack [70], tolerating faults across layers involving circuit design, firmware, operating system, applications, etc. Cross-layer fault tolerant systems have potential to implement reliable, high-performance and energy-efficient solutions without overwhelming costs [33] by distributing the responsibilities of tolerating faults across multiple layers [6]. Cross-layer Fault Tolerance has also been viewed from the perspective of sensor data layers [61].

In case Fault Detection and Fault Recovery are to be implemented in different system layers, then following challenges arise [72]:For statistical validation and metrics high confidence resource-light reliability and availability estimation is needed.Verification of resilience techniques, to be sure that resilience techniques perform under all possible scenarios.Reliability grades for testing and grading system-wide reliability and data integrity. Reliability may change under different workloads.

In addition to the cross-layer approach, a Multi-Layer approach [73] has also been proposed, where system layers are adapted to each other to reduce error propagation. However, in the opinion of the authors of the current paper, this does not constitute a principally distinct approach but, rather, an increment to the cross-layer approach.

## 8. Conclusions

The current paper presented a systematic survey on fault tolerant techniques in USNs and pointed out open research issues in this field. The paper considered fault tolerant techniques that are developed for underwater use or could be adapted for that. The techniques were divided into groups according to the taxonomy of Fault Tolerance tasks, and papers utilizing these techniques were discussed in sections corresponding to the tasks.

We collected top papers by conducting a systematic search from different online environments, related papers suggested by those environments, and sources cited by the collected papers. Next, we analyzed the collected papers, divided them into categories and discussed aspects covered in those papers. Areas of high research interest and open research issues in the scope of the initial criteria were detected and brought out. Additionally, in order to categorize and systematize the analyzed papers, taxonomies for fault sources and Fault Tolerance tasks were described, and a full table of the papers was presented.

The current paper is the first to investigate the state-of-the-art in Fault Tolerance, particularly cross-layer Fault Tolerance, in USNs. According to the survey, there is a lack of research covering the cross-layer Fault Tolerance aspect for underwater sensor networks. Therefore, the mentioned topic is a prospective candidate for future works on fault tolerant USNs.

## Figures and Tables

**Figure 1 sensors-21-03264-f001:**
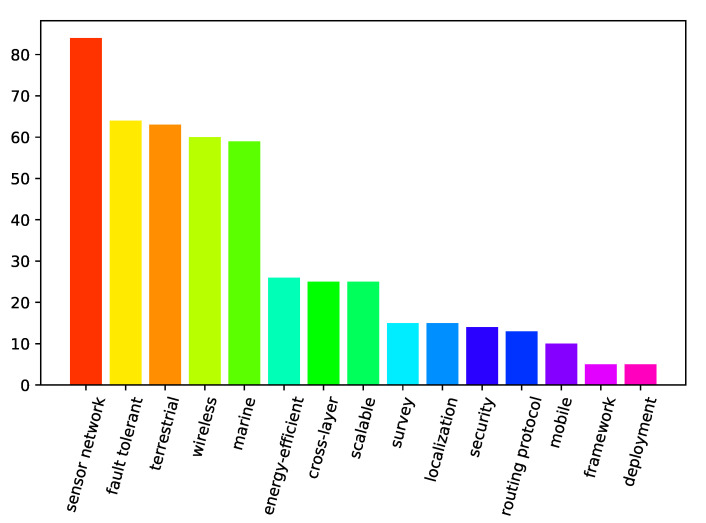
The number of papers by specific areas.

**Figure 2 sensors-21-03264-f002:**
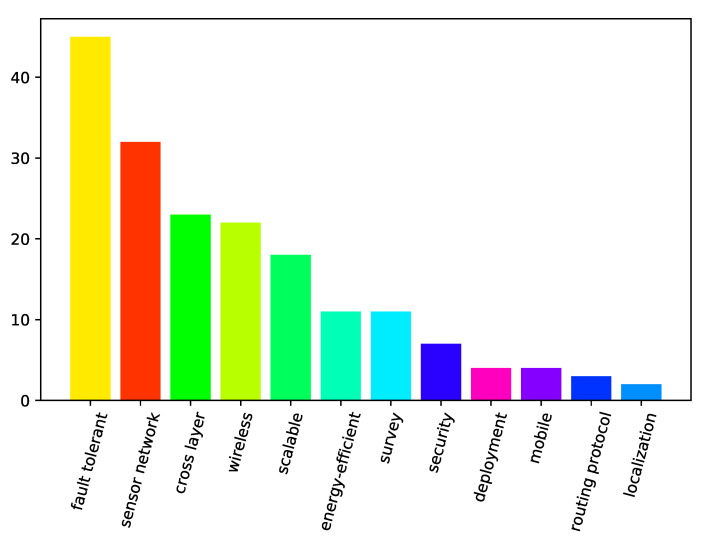
The number of terrestrial-related papers by specific areas.

**Figure 3 sensors-21-03264-f003:**
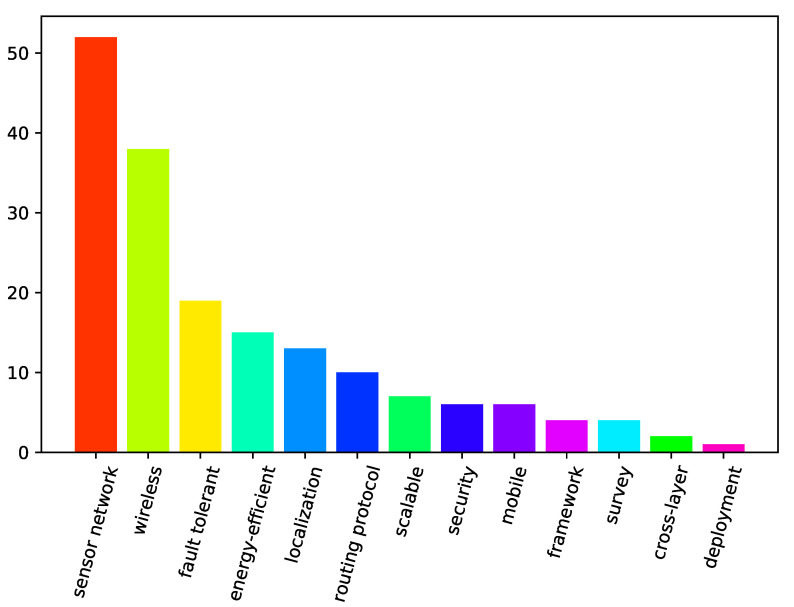
The number of marine-related papers by specific areas.

**Figure 4 sensors-21-03264-f004:**
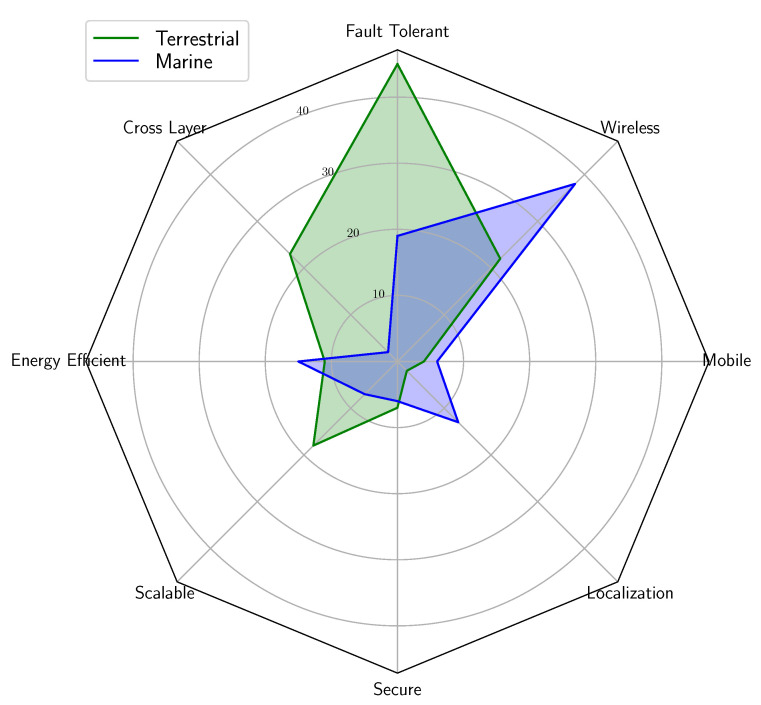
A radar chart of the analyzed papers addressing the main specific areas.

**Figure 5 sensors-21-03264-f005:**
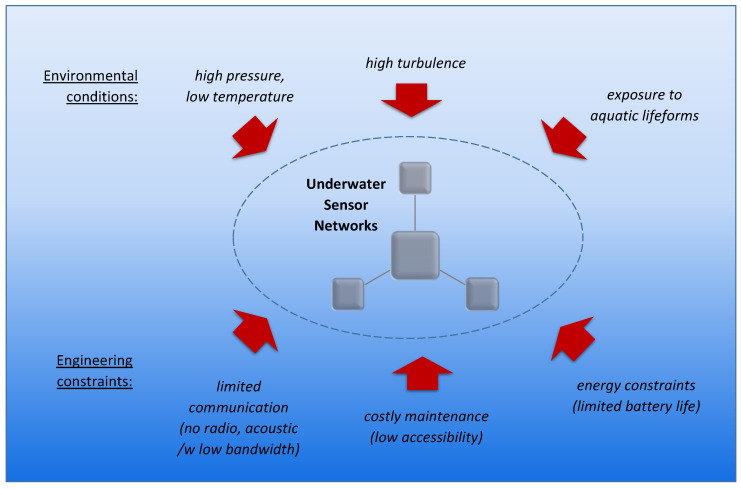
Environmental and engineering challenges in USNs.

**Figure 6 sensors-21-03264-f006:**
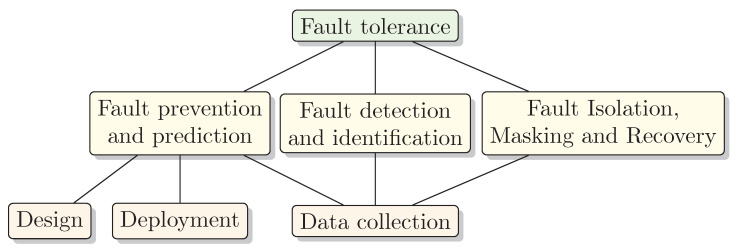
Taxonomy of Fault Tolerance tasks in USNs.

**Table 1 sensors-21-03264-t001:** Search engine result count of respective keyword combinations.

Search Keywords	G. Scholar	IEEEX	S.Direct
“underwater”, “internet of things”, “resilient”, “fault tolerant”, “fault management”, “cross-layer”	4	0	0
“underwater”, “internet of things”, “resilient”, “fault tolerant”, “fault management”	8	0	1
“underwater ”, “sensor network”, “resilient”, “fault tolerant”, “fault management”	36	0	4
“sensor network”, “resilient”, “fault tolerant”, “fault management”, “cross-layer”	49	1	4
“underwater ”, “sensor network”, “fault management”	162	0	10
“sensor network”, “resilient”, “fault tolerant”, “fault management”	223	9	16

**Table 2 sensors-21-03264-t002:** Categorized papers.

Pub.	1st Auth.	Year	Extra-Functional Aspect				Marine	Fault Tolerance (FT) Tasks and
			Secure	Energy-Efficient	Scalable	Cross-Layer		Other Research Areas
[1]	Domingo	2012	-	-	-	-	+	sensor network
[2]	Domingo	2012	-	-	-	+	+	sensor network, FT detect/recover, wireless
[3]	Zenia	2016	-	+	+	-	+	sensor network, routing protocol, survey, FT detect, FT recover
[4]	Jiang	2018	-	-	-	-	+	survey, wireless, sensor network
[5]	Kao	2017	-	-	-	-	+	FT design, survey, wireless
[6]	Veleski	2017	-	-	-	+	-	survey, FT detect, FT recover
[7]	Paradis	2007	-	-	-	+	-	FT detect/recover, survey, wireless
[8]	Mitra	2016	-	+	+	+	-	survey, wireless, FT detect, FT recover,
[9]	Atzori	2010	-	-	-	-	-	sensor network, survey
[10]	Diaz	2016	-	-	+	-	-	survey
[11]	Tyagi	2013	+	+	+	-	-	survey, wireless, routing protocol
[12]	Singh	2012	-	-	-	-	-	routing protocol, survey, wireless
[13]	Fadel	2015	-	-	-	-	-	survey, sensor network, wireless, FT detect
[14]	Moridi	2020	-	-	-	-	-	sensor network, wireless, FT detection, FT recovery
[15]	More	2017	-	+	+	-	-	sensor network, survey
[16]	Amin	2019	-	-	-	-	-	FT detect/recover, survey
[18]	Domingo	2009	-	-	-	-	+	FT detect, wireless
[19]	Xu	2012	-	+	+	-	+	FT detect, FT recover, sensor network
[20]	Lal	2016	+	-	+	-	+	wireless, sensor network
[21]	Das	2017	-	-	+	-	+	localization, sensor network, FT recover
[22]	Mohamed	2011	-	-	+	-	+	sensor network, FT detect
[23]	Kumar	2018	-	-	+	-	-	FT detect/recover
[25]	Khan	2013	-	-	+	+	-	FT detect/recover, wireless
[26]	Henkel	2011	-	-	-	+	-	FT design/detect/recover
[27]	Georgakos	2013	-	-	-	+	-	FT design/detect/recover, vehicle
[28]	Lorenz	2012	-	-	-	-	-	FT prevent
[29]	Sauli	2012	-	-	-	-	-	FT prevent
[30]	Rehman	2016	-	-	-	+	-	FT prevent/detect/recover
[31]	Kaaniche	2000	-	-	-	-	-	FT prevent/detect/recover
[33]	Carter	2010	-	-	-	+	-	FT design
[34]	Sofi	2018	-	+	-	-	+	sensor network, wireless
[35]	Noh	2016	-	-	-	-	+	routing protocol, sensor network, wireless
[37]	Wu	2007	-	-	-	-	-	deployment, localization, sensor network, wireless
[38]	Isler	2004	-	-	-	-	-	deployment, sensor network, wireless
[39]	Dong	2020	-	+	-	-	+	sensor network, wireless, FT recover
[40]	Cheng	2008	-	-	-	-	-	deployment, sensor network, wireless
[41]	Bhuvana	2018	-	+	-	-	-	sensor network, wireless, FT detect
[42]	Goyal	2017	-	-	-	-	+	wireless, sensor network
[43]	Goyal	2018	-	-	-	-	+	wireless, sensor network, FT detection, FT recovery
[45]	Campagnaro	2016	-	-	-	-	+	framework, wireless, sensor network
[46]	Petrioli	2015	-	-	-	-	+	framework wireless, sensor network
[47]	Petroccia	2013	-	-	-	-	+	framework, wireless, sensor network
[48]	Ateniese	2015	+	-	-	-	+	framework, wireless, sensor network
[49]	Liu	2013	+	+	-	+	-	wireless, sensor network
[51]	Gunawi	2011	-	-	+	-	-	FT design
[53]	Napolitano	1995	-	-	-	-	-	sensor network, FT detect, FT recover
[54]	Neti	1992	-	-	-	-	-	FT design
[55]	Benincasa	2014	-	-	-	-	-	sensor network
[56]	Compton	2012	-	-	-	-	-	sensor network, deployment
[57]	DAniello	2016	-	-	-	+	-	sensor network, FT detect, FT recover
[58]	Alansary	2019	-	-	-	-	+	vehicle, FT recovery
[59]	Cristea	2011	+	-	+	-	-	FT detect, FT recover
[60]	Salera	2007	-	-	-	+	-	sensor network, FT detect, FT recover
[61]	Vihman	2020	+	-	+	+	+	sensor network, FT detect
[62]	Han	2015	+	-	-	-	+	wireless, sensor network
[63]	Chae-Won	2016	+	-	-	-	+	sensor network, wireless
[64]	Han	2020	+	-	+	-	+	sensor network, wireless
[65]	Dong	2013	-	+	-	-	+	sensor network, wireless
[66]	Zhou	2016	-	+	-	-	+	, wireless, sensor network, routing protocol
[67]	Wang	2016	-	+	-	-	+	, sensor network, wireless
[68]	Huang	2011	-	+	-	-	+	wireless, sensor network, routing protocol
[69]	Rani	2017	-	+	-	-	+	sensor network, routing protocol
[70]	DeHon	2010	-	+	+	+	-	FT detect, FT recover
[71]	Darra	2017	+	-	-	-	-	survey, sensor network, wireless
[72]	Mitra	2010	-	-	-	+	-	FT detect, FT recover
[73]	Henkel	2014	-	-	-	+	-	FT detect, FT recover
[74]	Bulusu	2000	-	+	+	-	-	localization, sensor network
[75]	Nassif	2001	-	-	-	-	-	FT prevent
[76]	Zhao	2002	-	+	+	-	-	, wireless, sensor network
[77]	de Lemos	2004	-	-	-	-	-	FT design, sensor network
[78]	Bokareva	2005	-	-	-	+	-	cross-layer, FT design, FT recover, framework, sensor network
[79]	Heidemann	2006	-	-	-	-	+	sensor network, wireless
[80]	Mengjie	2007	-	-	+	+	-	wireless, sensor network, FT detect, FT recover
[81]	Lee	2008	-	-	-	-	-	wireless, FT detect, sensor network
[82]	Wang	2008	-	-	-	-	+	sensor network
[83]	Khan	2009	-	+	+	-	-	wireless, FT design, sensor network
[84]	Teymorian	2009	-	-	-	-	+	localization, sensor network
[85]	Yu	2009	-	-	-	-	+	localization, wireless, sensor network
[86]	Kim	2011	-	-	-	+	-	vehicle, FT detect, FT recover,
[87]	Tanasa	2011	-	-	-	-	-	vehicle, FT detect
[88]	Roman	2011	+	-	-	-	-	sensor network,
[89]	Paul	2011	+	-	-	-	-	sensor network
[90]	Xu	2011	-	-	-	-	+	wireless, sensor network, routing protocol, FT recovery
[91]	Thomas	2013	-	-	-	-	-	FT detect
[92]	Gubbi	2013	+	-	+	-	-	wireless, sensor network,
[93]	Guo	2013	-	-	+	-	+	localization, sensor network
[94]	Amory	2013	+	-	+	-	+	vehicle
[95]	Oteafy	2014	-	-	-	+	-	wireless, sensor network
[96]	Rault	2014	-	-	+	-	-	wireless, sensor network
[97]	Kuila	2014	-	+	-	-	-	wireless, sensor network, routing protocol
[98]	Zhu	2014	-	-	-	-	+	sensor network
[99]	Rossi	2015	-	+	-	-	+	sensor network, wireless
[100]	Bauer	2015	-	-	-	-	-	FT masking
[101]	Benson	2015	-	-	-	+	-	sensor network
[102]	Zhehao	2015	-	-	-	-	+	localization, wireless, sensor network
[103]	Han	2015	-	-	-	-	+	localization, wireless, sensor network, deployment
[104]	Valerio	2015	-	-	-	-	+	wireless, sensor network, routing protocol
[105]	Rehman	2016	-	-	-	+	-	FT detect, FT recover,
[106]	Sahoo	2016	-	-	-	+	-	FT design, FT detect
[107]	Li	2016	-	-	-	-	+	localization, vehicle
[108]	Liu	2016	-	-	-	-	+	sensor network, wireless, localization
[109]	Khan	2016	-	-	-	-	+	vehicle, sensor network
[110]	Koraz	2017	-	-	-	+	-	FT detect
[111]	Suvarna	2017	-	+	-	-	+	wireless, sensor network, routing protocol
[112]	Cario	2017	-	+	-	-	+	sensor network, wireless
[113]	Dong	2017	-	+	-	-	+	, localization, wireless, sensor network
[114]	Kao	2017	-	-	-	-	+	survey, sensor network, wireless
[115]	Mortazavi	2017	-	-	-	-	+	localization, wireless, sensor network
[116]	Seto	2017	-	-	-	-	+	vehicle
[117]	Azad	2018	-	-	-	+	-	FT detect, FT recovery
[118]	Sahu	2018	-	-	-	-	+	clustering, sensor network, routing protocol, FT detection, FT recovery
[119]	Dala	2018	-	-	-	-	+	sensor network, FT detection, FT recovery
[120]	Tang	2018	-	-	-	-	+	wireless, sensor network, fault, FT detection, FT recovery
[121]	Yanmaz	2018	-	-	+	-	-	vehicle, sensor network, wireless
[122]	Han	2018	-	-	-	-	+	localization, wireless, sensor network
[123]	Shah	2018	-	-	+	-	+	localization, sensor network
[124]	Caporuscio	2020	-	-	-	-	-	sensor network, FT detection, FT recovery
[125]	Desai	2020	-	-	-	-	-	sensor network, FT detection
[126]	Jin	2020	-	-	-	-	+	sensor network, wireless, routing protocol, vehicle, FT detection; FT recovery
[127]	Prasanth	2020	-	+	-	-	+	wireless, sensor network, fault, ft recovery, ft detection

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
