# Peer review of "Systematic Review of Fault Tolerant Techniques in Underwater Sensor Networks"

_sensors, 2021, doi:10.3390/s21093264_

Round 1

Reviewer 1 Report

In this manuscript, authors are proposing a Systematic Review of Fault Tolerant Techniques in Underwater Sensor Networks. 

This work doesn’t correspond to a systematic review. Several works concerning Fault Tolerant Techniques are referenced but there is not a comprehensive description of them. Moreover, many approaches that were mentioned are generic approaches and there is not a connection with Underwater Sensor Networks. Additionally, there is a lack of information about Underwater Sensor Networks, in terms of, for instance, task requirements, connectivity requirements, among others.

In terms of open issues, some of them were already addressed by other authors. It is not clear how the current approaches are failing.

Author Response

This work doesn’t correspond to a systematic review. Several works concerning Fault Tolerant Techniques are referenced but there is not a comprehensive description of them. 

Thank you for this comment. The objective of these survey was to generalize from the large pool of papers on USN (as well as some relevant generic) fault tolerance techniques. Since this pool includes 122 different works, there is not much space to go into details of the works. Therefore, the works were analyzed and grouped together according to different topics and aspects. In the new version of the paper we have attempted to improve the structure and general readability of the survey. 

Moreover, many approaches that were mentioned are generic approaches and there is not a connection with Underwater Sensor Networks. 

Thank you. This is a very relevant remark. As we responded to Review #4: Initial search for cross-layer fault tolerant USNs resulted in 4 matches as shown in the newly added Table 1. Of them only two were targeting underwater sensor networks. One of the results was the survey of underwater sensor network communication where 'cross-layer' didn't apply to system stack but only to communication protocol layers. Another was focusing on data-driven cross-layer fault tolerance in underwater sensor networks.

We expanded our initial search criteria to include terrestrial papers because of the small count of relevant underwater papers and because some of the fault tolerant techniques used in terrestrial environments are applicable in underwater environments.

This aspect has been elaborated now in Section 2.

Additionally, there is a lack of information about Underwater Sensor Networks, in terms of, for instance, task requirements, connectivity requirements, among others.

Thank you. this is a very valid remark. We have added a new Section 3 and a new Figure 5 to address this. We also updated the Introduction to explain this context.

In terms of open issues, some of them were already addressed by other authors. It is not clear how the current approaches are failing.

The existing approaches are not particularly failing. They are doing their job for a particular application scenario. However, there are gaps in terms of potentially more efficient/advanced concepts that could be implemented (e.g. cross-layer fault tolerance). Also, in some aspects only limited topics have been targeted. To clarify this situation, we have added more details in the comments of Table 2 and updated the description in Section 7.

Reviewer 2 Report

The topic is interesting and timely. The subject area is hot and aims at the readership of the journal. The structure is nice and clear. There are few issues though, as below:

  1. The introduction section is very short and does not introduce the work well. Please make sure to discuss the research problem in more details and list the contributions.
  2. Section 2 is nicely described but miss the formal methods. I would urge the authors to add formal description in order to improve the impact and quality of the work
  3. The IoT and energy efficiency which both deem essential in the paper have been briefly mentioned separately. You need to add further details about them with support of additional references. You may have a look at https://link.springer.com/article/10.1007%2Fs11227-020-03147-8
  4. In section 7, you may need to add the criteria that you used in the selection of these papers among others in the literature.
  5. Please add future work to Sec 9, and proofread the paper.

Author Response

1) The introduction section is very short and does not introduce the work well. Please make sure to discuss the research problem in more details and list the contributions.

We fully agree with this comment. We have now expanded the introduction by discussing the research problem in more detail, explaining the specifics of the underwater environment and providing the contributions of this paper.

2) Section 2 is nicely described but miss the formal methods. I would urge the authors to add formal description in order to improve the impact and quality of the work

We thank the reviewer for this comment. We have now expanded Section 2. Formal methodology for the search has been included along with the table with initial search results' statistics.

3) The IoT and energy efficiency which both deem essential in the paper have been briefly mentioned separately. You need to add further details about them with support of additional references. You may have a look at https://link.springer.com/article/10.1007%2Fs11227-020-03147-8

The recommended paper has been added to the section of open issues in energy-efficiency. furthermore, additional papers were identified and included to the analysis.

4) In section 7, you may need to add the criteria that you used in the selection of these papers among others in the literature.

A formal methodology description was added to Section 2 that explains the selection criteria. 

5) Please add future work to Sec 9, and proofread the paper.

Thank you for this comment. We have added a reference to the gaps and prospective topics for future work into the conclusions. The paper has been now thoroughly checked for typos.

Reviewer 3 Report

In this paper, the authors reviewed a group of cross-layer fault-tolerant techniques for UWSNs.

The introduction is not acceptable. It should be expanded by including a summary of the applications, problem definition, current solutions, research gap and the objective of the survey.

The fault tolerance tasks need to be explained in more details using some examples and figures. The advantages and disadvantages of these approaches need to be discussed.

Sections 4, 5, and 6 are too short and can be combined into one section.

Section 7 (Analysed papers) need to be expanded by generally discussing the qualitative comparison of these techniques (within table 1) in terms of security, energy, scalability and cross-layer capability.

I would suggest another proofreading as there are some grammatical errors and typos in this manuscript.

Author Response

* The introduction is not acceptable. It should be expanded by including a summary of the applications, problem definition, current solutions, research gap and the objective of the survey.

Thank you for this comment. We have now added all the mentioned missing points to the Introduction. See also the answer to question 1 of Reviewer 2. 

* The fault tolerance tasks need to be explained in more details using some examples and figures. The advantages and disadvantages of these approaches need to be discussed.

Thank you for this remark. We have now expanded Section 4.2 section, adding the explanation and presenting the advantages/disadvantages of each task.

* Sections 4, 5, and 6 are too short and can be combined into one section.

We agree with this suggestion. We have now combined Sections 4, 5 and 6 and added an introductory paragraph to the beginning of the new Section 4.

* Section 7 (Analysed papers) need to be expanded by generally discussing the qualitative comparison of these techniques (within table 1) in terms of security, energy, scalability and cross-layer capability.

Thank you for this comment. Section 6 Comparative Analysis has been added. In this section, qualitative comparisons of the analyzed papers are provided.

* I would suggest another proofreading as there are some grammatical errors and typos in this manuscript.

Thank you. Additional proofreading has been thoroughly executed and several mistakes fixed.

Reviewer 4 Report

This manuscript discusses the problem of fault detection and recovery of underwater acoustic network nodes. It has practical value and is an interesting problem. In general, the proposed scheme is not closely combined with the characteristics of underwater acoustic channels and underwater acoustic networks. The existing problems mainly include:

1) The introduction is too simple. The authors should add the literature on the overview of fault detection and recovery for underwater acoustic communication networks, and clearly point out the purpose, key points, ideas, and major contributions of this review. It is suggested that the authors refer to other review articles to improve the writing of this review.

2) It is suggested to add a Section to describe the characteristics of underwater acoustic communication and underwater acoustic network and point out the logical relationship between these characteristics and fault detection and repair, and then summarize the methods of fault detection and repair that have been adopted to solve these underwater acoustic characteristics in literature.

3) As a review paper, this paper has only five Figures, which is too few. It is suggested to add some logic diagrams to sort out the current work related to fault detection and repair in underwater acoustic networks and help readers to better clarify the relevant research progress. Suggest the authors rethink the writing frame of the paper.

4) The author's retrieval method is defective. For example, the following literature related to fault detection and recovery of underwater acoustic network nodes has not been retrieved:
---Jin Z, et al., "Routing Void Prediction and Repairing in AUV-Assisted Underwater Acoustic Sensor Networks," IEEE Access, vol. 8, no. 1, pp. 54200-54212, 2020.
---Y. Tang, et al., "Emergency Communication Schemes for Multi-hop Underwater Acoustic Cooperative Networks," in Proc. IEEE ICSPCC 2018, Qingdao, China,14-16 Sept. 2018.
---B. H. Khudayer, et al., "Efficient Route Discovery and Link Failure Detection Mechanisms for Source Routing Protocol in Mobile Ad-Hoc Networks," IEEE Access, vol. 8, pp. 24019-24032, 2020.
---N. Desai, et al., "Enhancing Fault Detection in Time Sensitive Networks using Machine Learning," in Proc. IEEE COMSNETS 2020, Bengaluru, India, 2020, pp. 714-719.

It is suggested that the authors further improve the literature retrieval strategy, modify the search terms, and supplement more references.

5) The authors should polish the language further.

Author Response

1) The introduction is too simple. The authors should add the literature on the overview of fault detection and recovery for underwater acoustic communication networks, and clearly point out the purpose, key points, ideas, and major contributions of this review. It is suggested that the authors refer to other review articles to improve the writing of this review.

Thank you for this comment. The Introduction has been now expanded considerably. We explain the key points, give an overview of related surveys and present the new contributions of this survey.

2) It is suggested to add a Section to describe the characteristics of underwater acoustic communication and underwater acoustic network and point out the logical relationship between these characteristics and fault detection and repair, and then summarize the methods of fault detection and repair that have been adopted to solve these underwater acoustic characteristics in literature.

A very valid and useful comment. We have added a new Section 3 and a new Figure 5 explaining the specifics and the distinct challenges of USNs.

3) As a review paper, this paper has only five Figures, which is too few. It is suggested to add some logic diagrams to sort out the current work related to fault detection and repair in underwater acoustic networks and help readers to better clarify the relevant research progress. Suggest the authors rethink the writing frame of the paper.

This is a very good and useful comment. We redesigned the structure and presentation of the paper and added visualisations. In particular, the following has been implemented:

  • We added new visualizations: included new Table 1, new Figure 5 and improved the readability of Table 2.
  • Structurewise, we introduced a new Section 3 and grouped the overview of techniques into a single section (Section 4).
  • We have also completely rewritten the Introduction to add the context and structure to the reader. We also elaborated the formal methodology followed in the systematic search.

4) The author's retrieval method is defective. For example, the following literature related to fault detection and recovery of underwater acoustic network nodes has not been retrieved:
---Jin Z, et al., "Routing Void Prediction and Repairing in AUV-Assisted Underwater Acoustic Sensor Networks," IEEE Access, vol. 8, no. 1, pp. 54200-54212, 2020.
---Y. Tang, et al., "Emergency Communication Schemes for Multi-hop Underwater Acoustic Cooperative Networks," in Proc. IEEE ICSPCC 2018, Qingdao, China,14-16 Sept. 2018.
---B. H. Khudayer, et al., "Efficient Route Discovery and Link Failure Detection Mechanisms for Source Routing Protocol in Mobile Ad-Hoc Networks," IEEE Access, vol. 8, pp. 24019-24032, 2020.
---N. Desai, et al., "Enhancing Fault Detection in Time Sensitive Networks using Machine Learning," in Proc. IEEE COMSNETS 2020, Bengaluru, India, 2020, pp. 714-719.

It is suggested that the authors further improve the literature retrieval strategy, modify the search terms, and supplement more references.

Thank you for pointing this out. The reason of not retrieving newer articles was that initial search was done in 2019 and was not repeated after newer papers got published and appeared in the databases. We conducted a new search and added new references.
 An article by B. H. Khudayer et al. that you brought out in your comment is not included in current review because it is about network topology and does not seem to cover any aspects that are specific for fault management in sensor networks or underwater environments.

5) The authors should polish the language further.

We have now thoroughly proofread the paper and fixed the language mistakes.

Reviewer 5 Report

The article is a detailed review of the available literature of fault tolerant techniques in sensor networks. The paper may be an important tool in the hand of researchers in the area, however I have some remarks and questions.

a) The authors use PRISMA guidelines for the review. Since those guidelines are designed for medical research, could the authors explain while are those significant for the current field?

b) Figure 1 to figure 4 are highly redundant. In my opinion, a figure like Figure 4. would be enough to describe the covered fields.

c) While the title and abstract focuses on underwater sensor networks, most of the analyzed articles are terrestrial.

d) Table 1. is hard to read. The table header should be repeated on every page. The table is sorted by citations, however the (actual) citation count is not showed. Any additional data (like first author, title) could also help the usage of the table. (Maybe the authors could provide an xls version of the table, that could be filtered based on the reader actual interest.)

Author Response

a) The authors use PRISMA guidelines for the review. Since those guidelines are designed for medical research, could the authors explain while are those significant for the current field?

The reason for basing on PRISMA was twofold. First, it was explicitly recommended by the journal. Second, the authors found it very useful as the PRISMA checklists allowed better design, structuring and presentation of the survey, while being generic enough and well applicable to the topic at hand.

b) Figure 1 to figure 4 are highly redundant. In my opinion, a figure like Figure 4. would be enough to describe the covered fields.

Thank you for this comment. While the information in these figures is indeed redundant they were meant to convey a slightly different message. Unfortunately, an explanation motivating these figures was missing in the initial submission. We have updated Section 2 with more discussion about the formal search and the purpose of the figures.

c) While the title and abstract focuses on underwater sensor networks, most of the analyzed articles are terrestrial.

Thank you. This is a very relevant remark. Initial search for cross-layer fault tolerant USNs resulted in 4 matches as shown in the newly added Table 1. Of them only two were targeting underwater sensor networks. One of the results was the survey of underwater sensor network communication where 'cross-layer' didn't apply to system stack but only to communication protocol layers. Another was focusing on data-driven cross-layer fault tolerance in underwater sensor networks.

We expanded our initial search criteria to include terrestrial papers because of the small count of relevant underwater papers and because some of the fault tolerant techniques used in terrestrial environments are applicable in underwater environments.

This aspect has been elaborated now in Section 2.

d) Table 1. is hard to read. The table header should be repeated on every page. The table is sorted by citations, however the (actual) citation count is not showed. Any additional data (like first author, title) could also help the usage of the table. (Maybe the authors could provide an xls version of the table, that could be filtered based on the reader actual interest.)

Thank you for your comment. The order was not by the citation count but by the citation order in current survey. We apologize for the confusion in the table's explanation. We added a new Comparative Analysis section with expanded description of this to the paper. We added a header to all the pages of the table and an additional column with the 1st author's surname.

We could generate an XLS or CSV table from this information. However, making it public needs to be agreed with the publisher.

Round 2

Reviewer 1 Report

This new version of the manuscript gives a more comprehensive view of the context and requirements for fault tolerant techniques used in Underwater Sensor Networks. However, it stills a generic survey with many papers listed, but without any deep analysis of them. It can be used as an introductory paper of the area.

Author Response

However, it stills a generic survey with many papers listed, but without any deep analysis of them.

 Thank you for your comment. MDPI Sensors Instructions for Authors specifically asks reviews to be concise -
 "Reviews: These provide concise and precise updates on the latest progress made in a given area of research. Systematic reviews should follow the PRISMA guidelines. The recommended length of a Review is more than 20 journal pages."

Reviewer 2 Report

The authors have addressed all my concerns and comments and the paper has been improved significantly. I have no other comments. 

Author Response

Thank you and many thanks for the comments that helped us improve the manuscript!

Reviewer 3 Report

Thanks to the authors who revised the paper. It has been improved and can be accepted. 

Author Response

(The authors gave the same response as above.)

Reviewer 4 Report

The authors have revised the manuscript according to the comments of reviewers and made a lot of progress. However, some of the content could be further improved. It is recommended to check carefully before publication. Thanks for the authors' work!

Author Response

However, some of the content could be further improved. It is recommended to check carefully before publication.

Thank you for your comment. We verified the consistency of the work further and fixed found errors in Table 2.

Reviewer 5 Report

The authors answered all my questions and I believe that the paper was significantly improved.

Author Response

Thank you and many thanks for the comments that helped us improve the manuscript!

This manuscript is a resubmission of an earlier submission. The following is a list of the peer review reports and author responses from that submission.